# Pregnancy in an *SRY*-Negative XX Developmental Sex Disorder Pig After Removing an Ovotestis

**DOI:** 10.3390/vetsci12030268

**Published:** 2025-03-13

**Authors:** Jewel Toenges, Ahmed Tibary, Jon Michael Oatley, Muhammad Salman Waqas, Colton Robbins, Michela Ciccarelli

**Affiliations:** 1Department of Veterinary Clinical Sciences, College of Veterinary Medicine, Washington State University, Pullman, WA 99164, USA; jewel.toenges@gmail.com (J.T.); tibary@wsu.edu (A.T.); salman.waqas@wsu.edu (M.S.W.) coltonrobbins73@gmail.com (C.R.); 2Center for Reproductive Biology, Washington State University, Pullman, WA 99164, USA; joatley@wsu.edu; 3School of Molecular Biosciences, College of Veterinary Medicine, Washington State University, Pullman, WA 99164, USA

**Keywords:** disorder of sex development, ovotestis, fertility, intersex, swine

## Abstract

Disorders of sex development (DSDs) in mammals are relatively common in veterinary medicine. They often present as complaints related to infertility and abnormal behavior in animals. Many incidences of DSDs occur when there is a mismatch between chromosomal gender (XY/XX) and gonadal development (testes/ovaries), and produce a range of phenotypic abnormalities. These conditions have historically been referred to as “hermaphroditism”. Reports of *SRY*-negative DSDs in domestic animals are infrequent. This case is notable as it is the first to describe surgical correction involving the removal of a unilateral ovotestis in a pig, who was then able to carry a normal pregnancy to term.

## 1. Introduction

Disorders of sexual differentiation (DSDs) in mammals are classified into chromosomal, gonadal, and phenotypical types. Due to infertility and the expenses associated with diagnosis and potential treatment, these abnormalities are common reproductive complaints in veterinary medicine. Sex reversal is a gonadal disorder of sex development characterized by a discrepancy between chromosomal gender (XY/XX) and gonadal development (testes/ovaries). In the earlier literature, it was also referred to as ‘hermaphroditism’ [1].

In mammals, the stage for sex determination is set at fertilization, but is induced during fetal gonadogenesis when the *SRY* gene is activated. Mammalian male (XY) zygotes express the *SRY* gene located on the Y chromosome, which initiates the process of testicular development. Sertoli cells in the newly formed testes produce Anti-Müllerian Hormone (AMH), which triggers the regression of the female paramesonephric ducts (Müllerian ducts) and promotes the differentiation of Leydig cells. As a result, testosterone and dihydrotestosterone are produced, stimulating the development of male reproductive ducts and external genitalia. In contrast, the absence of the Y chromosome in female (XX) zygotes leads to the formation of ovaries and the development of an internal and external female reproductive tract. In sex-reversed mammals, the presence or absence of the *SRY* gene or its mutations can lead to different abnormalities, such as XX *SRY*-negative masculinization and XY *SRY*-positive feminization. In domestic animals, sex reversal disorders have been described in pigs, goats, sheep, llamas, cattle, horses, cats, and dogs [2,3,4,5,6,7,8,9]. XX-*SRY*-negative masculinized individuals are particularly interesting because they might have an uncharacterized X chromosome or an autosomal gene that, when mutated, can induce testicular tissue development in the absence of the Y chromosome and *SRY* sequence. Several gene mutations could cause the affected phenotype. *SOX9* is the most involved gene [10]. In addition to *SOX9*, three other possible causes have been identified: *RSPO1* loss-of-function mutations, deletion of the genomic area, which encompasses the *FOXL2* gene promoter, and *SOX3* duplications. Various candidate genes have been ruled out as explanations for the XX male phenotype that is *SRY* negative observed in dogs, including *PISRT1*, *RSPO1*, *SOX9*, *GATA4*, *FOG2*, *LHX1*, *DMRT1*, *LHX9*, and *WT1* [11,12,13,14].

Most XX masculinized animals are phenotypically female, but some can show different degrees of virilization, ranging from abnormal vulvar conformation (“skyhook”) to the presence of the epididymis and vas deferens. Similar cases were reported in goats and pigs as an inherited autosomal recessive syndrome [1]. Pregnancies from animals with DSDs have not been reported in the peer-reviewed scientific literature. The incidence of XX *SRY*-negative DSDs in swine is relatively higher than in other species, with 0.1 to 0.6% of the population affected [2]. Little is known about the root of this disorder. Here, we describe a case of an *SRY*-negative XX DSD with partial masculinization in a gilt who was able to carry a pregnancy to term after unilateral gonadectomy (an ovotestis). Pigs serve as significant biomedical models for human diseases. The ability of this sow to regain fertility after unilateral gonadectomy of an ovotestis could be important for women with the same DSD. In humans, the incidence of this type of DSD is 1/20,000 [15]. Most (90%) of the cases are *SRY* + due to *SRY* translocation and Y chromosome chimerism. There are 11 reported cases of pregnancy in humans diagnosed with true hermaphroditism, but none underwent advanced genetic testing. In these cases, there is a higher incidence of male progeny. However, the genetic basis behind this is unknown at this time [16].

## 2. Detailed Case Description

A 2-year-old Large White research gilt was presented to the Comparative Theriogenology service at WSU for infertility. She was delivered via cesarean section in the Spring of 2019, following the transfer of embryos that were generated via in vitro fertilization and electroporated with CRISPR-Cas9 reagents designed to target the *NANOS2* gene. The gilt failed to become pregnant after several attempts at artificial insemination. She was a mosaic for the *NANOS2* gene involved in an unrelated research project at WSU. Previous studies demonstrated that animals with inactivated alleles for the *NANOS2* gene have male-specific sterility due to the neonatal apoptosis of prospermatogonia, but females retain germlines and are fertile [17]. The first estrus was observed at 5 months of age, but was accompanied by aggressive, boar-like behavior. Due to the importance of piglets carrying the mutation of this gilt, artificial insemination was attempted on five occasions after estrus control using altrenogest (Matrix^®^, Jersey City, NJ, USA) (14 days treatment, 20 mg PO/day). No pregnancy was established after all these inseminations despite the use of satisfactory semen (concentration, motility, and morphology via a Computer-Assisted Semen Analysis (CASA) from a proven fertile boar. Interestingly, the insemination pipette could only be advanced 10 cm into the vagina, which is considerably shorter than the average 25 cm penetration in normal gilts [18]. During the physical examination, the vital parameters were within the normal limits. The only abnormalities observed were an abnormal vulvar conformation (“skyhook” vulva) (Figure 1), a typical external phenotype in gilts affected by DSDs, an excessive stature, and aggressive behavior. Transabdominal ultrasonography revealed a normal uterus and a left ovary with multiple follicles. The right gonad had an unusual echotexture, including a testis-like round structure measuring approximately 4 cm and an adjacent small portion of ovary-like tissue with small follicles (Figure 2). A DSD was suspected, given the animal phenotype and ultrasonographic findings. Karyotyping and a polymerase chain reaction for the *SRY*-gene were then conducted. The gilt’s genotype was confirmed to be chromosomally normal as 38, XX, and SRY-negative.

### 2.1. Surgical Procedure

An exploratory laparotomy was scheduled for two weeks after the diagnostic workup. Anesthesia was induced with xylazine (AnaSed LA; VetOne Inc., Boise, ID, USA; 3 mg/kg, IV), ketamine (Ketaset; Zoetis Inc., Parsippany, NJ, USA; 5–10 mg/kg, IV), propofol (Propoflo; Zoetis Inc., Parsippany, NJ, USA; 3 mg/kg titrated, IV), and maintained with isoflurane in O_2_. Ceftiofur crystalline free acid (EXCEDE^®^; Zoetis Inc., Parsippany, NJ, USA; 6.6 mg/kg, SQ) and flunixin meglumine (Prevail; VetOne Inc., Boise, ID, USA; 1.1 mg/kg, IV) were administered during the surgery. A standard midline laparotomy was performed. Both gonads and the uterus were exteriorized and evaluated (Figure 3). The uterine horns were grossly normal. The left gonad has a normal ovary appearance with twelve corpora lutea of 5 to 8 mm in diameter. The right gonad consisted of two morphologically distinct parts: one round, smooth mass of 4 cm in size, and the other with four corpora lutea of 4–8 mm in diameter (Figure 3). The abnormal gonad was attached to the right uterine horn by a 1 to 2 cm wide and 6 cm long tubular structure resembling an epididymis. Tortuous vessels were attached to the proximal end of the epididymis-like structure and appeared like the pampiniform plexus. Intra-operatively, the decision was made to excise the abnormal right gonad. The ovarian pedicle was freed from the broad ligament and ligated using the 3-clump approach and a combination of encircling and transfixing ligatures of No. 0 polydioxanone suture (PDS (R)II; Ethicon Inc., Raritan, NJ, USA). The uterus and left ovary were replaced into the abdominal cavity, and the incision was routinely closed. The pig recovered uneventfully and was returned to the research facility. She received meloxicam (meloxicam tablets; Zydus Pharmaceuticals Inc., Solana Beach, CA, USA; 0.4 mg/kg PO q24) to reduce the pain from surgery.

### 2.2. Histopathology

The excised gonad was fixed in Buoin’s solution and processed for paraffin embedding, and cross-sections were stained with hematoxylin and eosin. The smaller ovary-like portion consisted of luteal tissue characterized by large, polygonal cells with abundant vacuolated cytoplasm and granulosa lutein cells, surrounded by a layer of smaller theca lutein cells, and small follicles presenting the typical fluid-filled cavity, surrounded by granulosa cells, which are further encircled by a layer of theca cells, all embedded within the ovarian stroma. Histology of the testicle-like structure revealed seminiferous tubules lacking spermatogenesis and showing atrophy and epithelial degeneration, with a disorganized arrangement of Sertoli cells (Figure 4). Based on the histopathology and cytogenetic results, the gilt was confirmed to be an XX *SRY*-negative DSD pig with a unilateral ovotestis.

### 2.3. Clinical Outcome

The gilt showed her first clinical signs of estrus, characterized by a complete immobility reflex, swelling of the vulva, and receptivity to a teaser boar, 4 months post-operatively. She was artificially inseminated with fresh semen collected from the same boar used initially. Pregnancy was confirmed via transabdominal ultrasonography 28 days post insemination. The pregnancy was monitored periodically via ultrasonography throughout the gestation. The gilt delivered six piglets without intervention following a normal gestation of 114 days. The litter contained four females and two males, none of which demonstrated gross signs of disorders of sexual development.

At the end of the research project, the sow was humanely euthanized. Upon necropsy, the right uterine horn presented cystic endometrial hyperplasia (CEH) and mucometra (8 L of mucus were collected), and a complete septum distal to the bifurcation separating the right uterine horn from the left (Figure 5 and Figure 6). The left ovary and uterine horn were grossly and histologically normal. Histological evaluation of the epididymis-like structure identified at the end of the right uterine horn could not distinguish between the oviduct or epididymis due to the similar cellular composition of both tissues. It was concluded that the complete septum allowed the pregnancy to occur only in the left horn. The unilateral right mucometra probably developed due to prolonged exposure to progesterone during pregnancy.

In summary, the gilt was a 38, XX *SRY*-negative DSD pig who returned to fertility after the excision of the right ovotestis. This gilt maintained a normal gestation and parturition in the left uterine horn due to a septum blocking the right uterine horn. The phenotypic abnormalities identified included a “skyhook” vulva, short vagina, a right ovotestis, epididymis-like structure, the complete uterine septum of the right uterine horn, unilateral right cystic endometrial hyperplasia (CEH), and mucometra.

## 3. Discussion

XX *SRY*-negative, with masculinization, is a disorder of sex development (DSD) characterized by a partial mismatch between the chromosomic and the gonadal sex. Affected individuals commonly show female phenotypes with primary anestrus and abnormal vulvar conformation or bilateral cryptorchidism and abnormal prepuce and penis. The gonads are bilateral ovotestes in most cases; a few instances have an ovotestis and an ovary, with one ovotestis and one testis being the least common. The phenotypical masculinization depends on the amount of testicular tissue present and serum testosterone levels. The animals affected by this disorder are sterile [1].

This case demonstrates several unique features of gonadal DSD. One of the most important findings was the ability of the gilt to return to fertility after the excision of the ovotestis. There are currently no reports of pregnancy in pigs with DSDs after the surgical removal of the abnormal gonadal tissue. There is, however, one report published in 1964 of pregnancy in a sow with one ovary and one testis. However, no cytogenetic or endocrine studies were performed [19]. The infertility of the gilt presented here, before unilateral gonadectomy, was most likely due to the effects of testosterone, generating negative feedback on the estrous cycle. This case demonstrates the feasibility of achieving pregnancy in an animal with gonadal disorders of sexual development. However, this requires at least one functional ovary, uterine body, and cervix. The surgical correction of pigs with masculinizing DSDs may help alleviate some of the economic losses of producers due to the infertility of gilts, aggressive behaviors, and boar taint from ovo-testicular tissue, and provide a model for other species. Although no similar cases were reported in pigs, the removal of the ovotestis and recovery of cyclicity was published in humans. In a study by Greeley et al. [20], a neonate was diagnosed as 46, XX *SRY*-negative sex reversal. This patient’s ovotestis produced insufficient testosterone to fully masculinize the genitalia, which accounts for the ambiguity of the external genitalia and persistence of incomplete Wolffian duct derivatives, and produced insufficient AMH to inhibit the formation of the Müllerian system, which accounts for the presence of a normal uterus and fallopian tubes. Additionally, ovarian follicular development was inhibited secondary to the presence of testicular tissue. After ovotestis removal at 3 weeks of age, serum AMH levels became low within a month, but the elevated testosterone was slow to resolve. Ovarian morphology and function gradually normalized as neonatal mini puberty waned.

Interestingly, evidence of a similar process [20] was observed in the case presented here, particularly regarding the pig phenotype. This included the presence of a “skyhook” vulva and the development of a uterus. However, abdominal ultrasonography and exploratory laparotomy showed that dominant follicles could develop, and that ovulation occurred. We theorized that elevated levels of systemic testosterone likely caused the fertility issues experienced by this gilt. This hormonal imbalance may interfere with the critical process of embryonic movement in the oviduct to enter the uterus and embryonic implantation, potentially disrupting the conditions necessary for a successful pregnancy. Post-operatively, the gilt achieved a normal estrus, immobility reflex, and ovulation, which resulted in pregnancy. This is important for future cases in pigs and humans and suggests a similar physiologic pattern in other mammalian species with XX *SRY*-negative DSDs.

The prevalence of DSDs in pigs is 0.1–0.6%, and is hypothesized to be sporadic or familial in inheritance, with suspicion of autosomal recessive mutation [2]. These genetic mutations have yet to be identified in pigs, but certain sires have been reported to produce 4–5% of offspring with DSDs [21]. There is no information on the prevalence or inheritance pattern of DSDs through maternal lines. Further investigation, potentially with the propagation of sire/dam lines with an increased incidence of DSDs, could contribute to identifying the genetics that play an essential role in sex development and differentiation. A good model for researching the autosomal gene or mutation is the Polled Intersex Syndrome (PIS) seen in goats. In the case of PIS, individuals with a mutated gene on chromosome 1q34 are associated with intersexual development, despite the absence of *SRY* [22]. Understanding the genetics behind DSDs will not only help prevent their adverse effects on industry, but potentially better explain these disorders in other species, possibly humans.

Importantly, this case demonstrates the possibility of the formation of testicular tissue in the absence of *SRY*, which is currently theorized as the initiating gene for male differentiation. Histological evaluation of the ovotestis demonstrated normal testicular parenchyma and a lack of the spermatogenic lineage, consistent with analog cryptorchidism in males and other reports on swine DSDs [21,23,24]. Additional reports of XX *SRY*-negative DSDs with masculinization have been described in dogs, goats, and horses [2,25,26,27,28]. In a study by Wertz and Herrman [29], twenty-one genes were described to have a dimorphic expression in the gonad, further complicating sexual differentiation and the current understanding of its regulatory mechanisms. Therefore, sexual differentiation requires further investigation to identify other genes that play a critical and integrative role in male or female sexual development.

Several interesting features should be highlighted in this case, including complete uterine septum, unilateral cystic endometrial hyperplasia (CEH), and mucometra. During normal sexual development, the Müllerian ducts fuse caudomedially, and the septum that separates the two horns regresses to create a tubular connection between the uterine horns. Lack of regression of the septum can lead to a ‘complete septate uterus’, also known as Robert’s uterus, a condition classified as Va by the American Society of Reproductive Medicine in human medicine [30]. These anomalies are also commonly 40% associated with renal anomalies [31]. No renal anomalies were observed in the subject of this case report. In humans, this complete uterine septum obstructs menstrual flow in one cavity, resulting in hematometra, hematosalpinx, and sometimes endometriosis [32]. As a consequence of the complete septum, the gilt developed mucometra, which mirrors hematometra in humans. Upon necropsy, unilateral Cystic Endometrial Hyperplasia (CEH) was observed in the right uterine horn. CEH is a disorder of the endometrial lining of unusual proliferation, hyperplasia of the endometrial glands, and stroma. Cystic endometrial hyperplasia has been documented in older (>3 years old) domesticated pet pigs, but rarely in commercial breeds. The lack of reports of CEH in commercial pigs is likely due to the nature of swine production, slaughter age, and average production life span. The incidence of CEH has been increasing in the UK, most likely due to the popularity of pet pigs and better swine medicine and care [33]. The etiology of CEH in humans has been described as being due to hyperestrogenism, relating to treatment with estrogens without progesterone [34]. In pigs, CEH has been attributed to hyperestrogenism, as seen in a study using zearalenone, an estrogen-producing mycotoxin, which induced CEH [35]. These observations by Wood et al., 2020 [33] suggest that pet pigs may experience an incidence of CEH due to increased estrogens from consistent estrous cycles throughout their lifespan.

The sow showed no clinical signs of mucometra; the pathology was only recognized during necropsy. There was no obvious evidence of this uterine disorder during the exploratory laparotomy in October 2020, possibly because it was in its initial stage. It was speculated that the reason why only the right uterine horn was affected by CEH was that it did not carry a pregnancy, and it was exposed to estrogens, testosterone, and progesterone for several years.

## 4. Conclusions

The presented case is novel in its presentation and subsequent treatment. Before this case, pregnancy achieved after the removal of a unilateral ovotestis had never been reported in animals. In humans, 11 cases were described; however, they lacked advanced genetic testing [16]. To our knowledge, our case represents the only confirmed case of a XX *SRY*-negative DSD pig with a complete uterine septum that led to unilateral CEH and mucometra.

## Figures and Tables

**Figure 1 vetsci-12-00268-f001:**
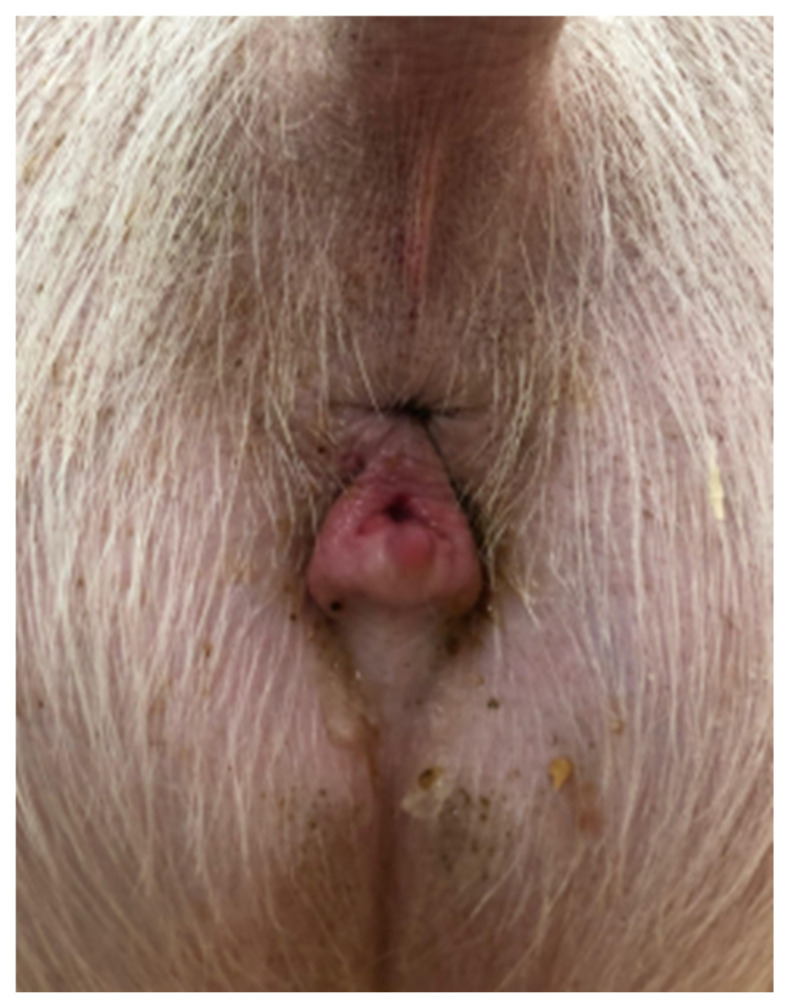
“Skyhook” conformation of the sow vulva.

**Figure 2 vetsci-12-00268-f002:**
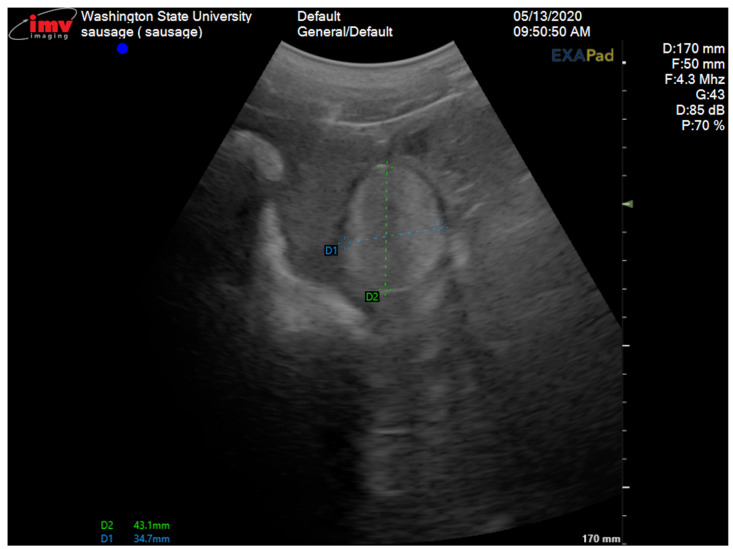
Transabdominal ultrasonogram of the right gonad showing the testicular tissue area measuring 43.1 mm × 34.7 mm.

**Figure 3 vetsci-12-00268-f003:**
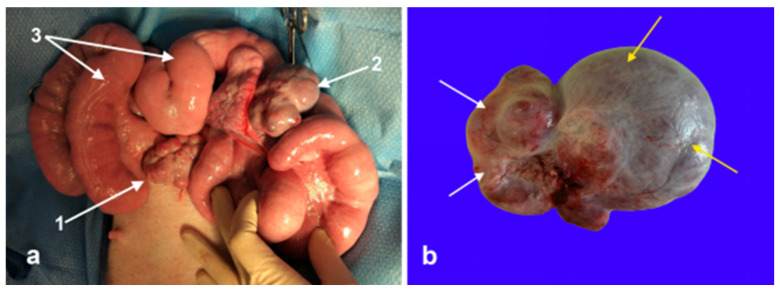
(**a**) Intra-operative image of the reproductive tract before surgical excision of the ovotestis. 1: Left normal ovary, 2: ovotestis, and 3: uterine horns. (**b**) Excised ovotestis, the ovarian tissue, and testicular tissue are indicated by the white and yellow arrows, respectively.

**Figure 4 vetsci-12-00268-f004:**
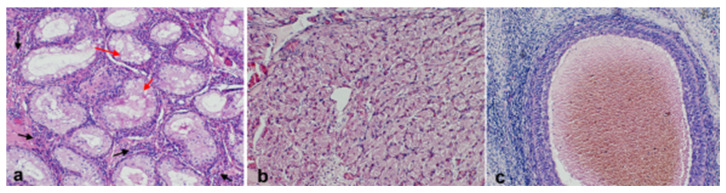
Photomicrographs of the histological section of the ovotestis. (**a**) Testicular tissue with Leydig cells with the interstitial tissue (black arrows) and seminiferous tubules with vacuolated Sertoli cells (red arrows); (**b**) normal luteal tissue; and (**c**) follicle.

**Figure 5 vetsci-12-00268-f005:**
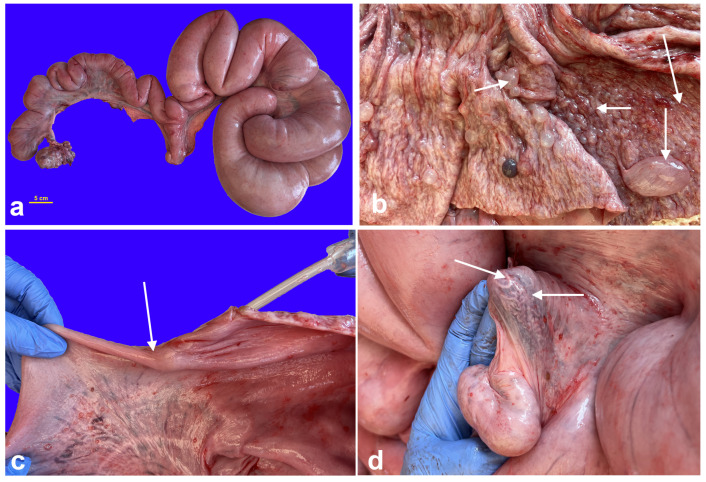
Images of necropsy findings. (**a**) Reproductive tract in toto, note the enlarged right uterine horn (mucometra), (**b**) cystic endometrial hyperplasia within the right uterine horn (arrows indicate cysts), (**c**) complete septum separating the two uterine horns (arrow), a pipette was inserted in the uterine lumen to demonstrate the occlusion, and (**d**) spermatic cord-like and epididymis-like structures (arrows) at the tip of the right uterine horn.

**Figure 6 vetsci-12-00268-f006:**
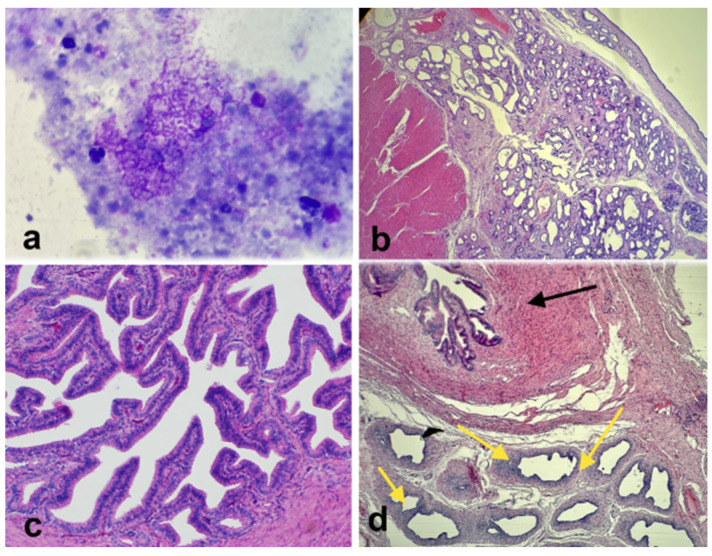
(**a**) Photomicrograph of cytological evaluation of the intra-uterine fluid showing no bacteria. (**b**) Histological section of the right uterine horn showing significant hyperplasia and endometrial cysts. (**c**) Histological section of the normal left oviduct. (**d**) Histological section of the abnormal right side shows oviductal tissue (black arrow) and epididymis-like tissue (yellow arrows).

## Data Availability

The original contributions presented in this study are included in the article. Further inquiries can be directed to the corresponding author.

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
