# Peer review of "Pregnancy in an *SRY*-Negative XX Developmental Sex Disorder Pig After Removing an Ovotestis"

_vetsci, 2025, doi:10.3390/vetsci12030268_

Round 1
Reviewer 1 Report
Comments and Suggestions for Authors
This research report is well written and scientifically sound. It contributes to the field as a clear explanation of disorders of sex development in pigs that may be comparable in other domestic species and therefore is of economic importance as the authors correctly suggest by suggesting treatment of DSDby surgery.
Author Response
Comment: This research report is well written and scientifically sound. It contributes to the field as a clear explanation of disorders of sex development in pigs that may be comparable in other domestic species and, therefore, is of economic importance as the authors correctly suggest by suggesting treatment of DSD by surgery.
Response: Thank you for your review. No response is needed.
Reviewer 2 Report
Comments and Suggestions for Authors
This interesting case describes a disorder of sex development (DSD) in a 2-year-old SRY-negative gilt that was created by the transfer of genetically modified embryos from in vitro fertilization. CRISPR-Cas9 was used to mutate the NANOS2 gene, which promotes male germ cell differentiation and simultaneously suppresses female fate pathways.
Surgical removal of a unilateral ovotestis successfully restored fertility and enabled the gilt to conceive and deliver six piglets, emphasizing the effectiveness of the surgical procedure in similar cases. The case is well presented, with only a few minor points raised.
- Were serum testosterone levels measured at any time before the removal of the ovotestis and after the surgery?
- The possible effect of CRISPR-Cas9 manipulation to mutate the NANOS2 gene on the presented SRY-negative DSD case was not discussed. Could there be a potential causal link?
- Page 4. Figure 2 legend.
Figure 2. Transabdominal ultrasonogram of the left gonad showing the testicular tissue area measuring 43.1 mm x 34.7 mm.
As a testis-like structure was identified on the right gonad, the Figure 2 legend should be corrected to: Transabdominal ultrasonogram of the right gonad showing the testicular tissue area measuring 43.1 mm x 34.7 mm.
- Page 5. Figure 4. Photomicrographs of the histological section of ovotestis.
Figure 4a. The testicular portion of the ovotestis resembles a male cryptorchid testis, particularly vacuolated Sertoli cells. The morphology of the Leydig cells, based on the current image, does not suggest active testosterone synthesis. Please comment on this observation or provide a higher magnification photomicrograph of the interstitial compartment for further evaluation.
Author Response
Comment 1: Were serum testosterone levels measured at any time before the removal of the ovotestis and after the surgery?
Response 1: The evaluation of testosterone levels was proposed to the Principal Investigator (PI), but it was denied. Additionally, this information was not necessary for the treatment. However, the gilt was taller and leaner than usual and displayed boar-like behavior.
Comment 2: The possible effect of CRISPR-Cas9 manipulation to mutate the NANOS2 gene on the presented SRY-negative DSD case was not discussed. Could there be a potential causal link?
Response 2: The homozygous NANOS2 knockout males are sterile due to neonatal apoptosis of prospermatogonia. The heterozygous males and females and homozygous knockout females retain a germline and are fertile.1 This is supported by the work of Park et al., who additionally demonstrated that boars with a homozygous deletion of NANOS2 are sterile, while homozygous knockout gilts are fertile.2 In the study by Park et al., homozygous knockout gilts achieved puberty around 5-8 months, showed normal signs of estrus, were bred via artificial insemination, and were diagnosed as pregnant at 28 days of gestation via ultrasonography. On histopathology of the ovaries, there was normal folliculogenesis. Notably, the expression of NANOS2occurs after sexual differentiation. Therefore, it does not play a role in disorders of sexual development. This evidence makes the subject of this case study’s disorder of sexual development less likely related to her genetic modification.
- Tsuda M. et al. (2003).Conserved role of nanos proteins in germ cell development. Science301, 1239–1241, doi: 10.1126/science.1085222.
- Park K.E., Kaucher A.V., Powell A., Waqas M.S., Sandmaier S.E., Oatley M.J., Park C.H., Tibary A., Donovan D.M., Blomberg L.A., Lillico S.G., Whitelaw C.B., Mileham A., Telugu B.P., Oatley J.M. (2017) Generation of germline ablated male pigs by CRISPR/Cas9 editing of the NANOS2 gene. Sci Rep. 7:40176. doi: 10.1038/srep40176. PMID: 28071690; PMCID: PMC5223215.
Comment 3: Page 4. Figure 2 legend.
Figure 2. Transabdominal ultrasonogram of the left gonad showing the testicular tissue area measuring 43.1 mm x 34.7 mm.
As a testis-like structure was identified on the right gonad, the Figure 2 legend should be corrected to: Transabdominal ultrasonogram of the right gonad showing the testicular tissue area measuring 43.1 mm x 34.7 mm.
Response 3: The correction was made. See manuscript.
Comment 4: Page 5. Figure 4. Photomicrographs of the histological section of ovotestis.
Figure 4a. The testicular portion of the ovotestis resembles a male cryptorchid testis, particularly vacuolated Sertoli cells. The morphology of the Leydig cells, based on the current image, does not suggest active testosterone synthesis. Please comment on this observation or provide a higher magnification photomicrograph of the interstitial compartment for further evaluation.
Response 4: After consulting with multiple board-certified pathologists, we conclude that we cannot make definitive statements regarding testosterone production from Leydig cells based solely on histomorphology. However, we do agree with the reviewer that a higher amount of vacuolated cytoplasm is correlated with an increased cytoplasmic content of steroid (lipid-soluble) hormones.
Reviewer 3 Report
Comments and Suggestions for Authors
The introduction gives a good background and covers the key references needed to understand the study, which describes a sow with a disorder in sexual development that rendered it sterile, but after undergoing surgery, it was able to carry out a pregnancy. The research design fits well with the goals, and results are straightforward and easy to see what’s going on. The conclusions make sense based on the results and tie everything together nicely. Overall, the study is well put together and gets the job done, however, it's a shame not to include the NANOS genotype information, as I believe it could provide valuable insights for the scientific community.
The study mentions that the sow was originated by CRISPR editing of the NANOS gene in zygotes. It is also noted that the sow is mosaic. Unfortunately, the study does not include a description of the NANOS gene alleles in this sow, specifically the alleles present in the ovotestis. Given the phenotypic background of mutations in the NANOS gene, something the authors are aware of and mention in the introduction, a probable hypothesis is that a new NANOS allele generated by CRISPR is responsible for the observed phenotype in the sow. The sexual reversal of SRY negative XX is of great interest, in order to improve the knowledge in sexual development pathway and it is possible that the NANOS edits have generated a highly interesting allele.
Author Response
Comment 1: The study mentions that the sow was originated by CRISPR editing of the NANOS gene in zygotes. It is also noted that the sow is mosaic. Unfortunately, the study does not include a description of the NANOS gene alleles in this sow, specifically the alleles present in the ovotestis. Given the phenotypic background of mutations in the NANOS gene, something the authors are aware of and mention in the introduction, a probable hypothesis is that a new NANOS allele generated by CRISPR is responsible for the observed phenotype in the sow. The sexual reversal of SRY negative XX is of great interest, in order to improve the knowledge in sexual development pathway and it is possible that the NANOS edits have generated a highly interesting allele.
Response1: We did not evaluate the alleles present in the ovotestis because we ruled out the possibility that the NANOS edits created a different allele; if that had been the case, both gonads would have exhibited abnormalities. In contrast, the ovary successfully developed follicles, ovulated, and maintained a pregnancy with corpora lutea.